# Nanoplastics and Arsenic Co-Exposures Exacerbate Oncogenic Biomarkers under an In Vitro Long-Term Exposure Scenario

**DOI:** 10.3390/ijms23062958

**Published:** 2022-03-09

**Authors:** Irene Barguilla, Josefa Domenech, Laura Rubio, Ricard Marcos, Alba Hernández

**Affiliations:** 1Group of Mutagenesis, Department of Genetics and Microbiology, Faculty of Biosciences, Universitat Autònoma de Barcelona, 08193 Cerdanyola del Vallès, Spain; irene.barguilla-moreno@lyon.unicancer.fr (I.B.); josefa.domenech@ttl.fi (J.D.); 2Nanobiology Laboratory, Department of Natural and Exact Sciences, Pontificia Universidad Católica Madre y Maestra (PUCMM), Santiago de los Caballeros 51000, Dominican Republic; l.rubio@ce.pucmm.edu.do

**Keywords:** polystyrene nanoplastics, arsenic, long-term co-exposure, cell transformation, DNA damage, oncogenic phenotype, carcinogenesis

## Abstract

The increasing accumulation of plastic waste and the widespread presence of its derivatives, micro- and nanoplastics (MNPLs), call for an urgent evaluation of their potential health risks. In the environment, MNPLs coexist with other known hazardous contaminants and, thus, an interesting question arises as to whether MNPLs can act as carriers of such pollutants, modulating their uptake and their harmful effects. In this context, we have examined the interaction and joint effects of two relevant water contaminants: arsenic and polystyrene nanoplastics (PSNPLs), the latter being a model of nanoplastics. Since both agents are persistent pollutants, their potential effects have been evaluated under a chronic exposure scenario and measuring different effect biomarkers involved in the cell transformation process. Mouse embryonic fibroblasts deficient for oxidative DNA damage repair mechanisms, and showing a cell transformation status, were used as a sensitive cell model. Such cells were exposed to PSNPLs, arsenic, and a combination PSNPLs/arsenic for 12 weeks. Interestingly, a physical interaction between both pollutants was demonstrated by using TEM/EDX methodologies. Results also indicate that the continuous co-exposure enhances the DNA damage and the aggressive features of the initially transformed phenotype. Remarkably, co-exposed cells present a higher proportion of spindle-like cells within the population, an increased capacity to grow independently of anchorage, as well as enhanced migrating and invading potential when compared to cells exposed to arsenic or PSNPLs alone. This study highlights the need for further studies exploring the long-term effects of contaminants of emerging concern, such as MNPLs, and the importance of considering the behavior of mixtures as part of the hazard and human risk assessment approaches.

## 1. Introduction

The generalized utilization of single-use plastic in uncountable applications comes together with an alarming rise in the levels of plastic waste, which is becoming a pressing environmental issue. Under environmental conditions, plastic litter is subjected to fragmentation and degradation into the so-called micro- and nanoplastics (MNPLs) [1,2]. While macroplastic pollution is the noticeable facet of this issue, the widespread presence of MNPLs in water, soil, and the air is equally worrisome. Indeed, it has recently been estimated that 8.3 million MNPLs particles contaminate each cubic meter of ocean water [3], and they are ubiquitously distributed in open seas and coasts [4]. Moreover, current research indicates that MNPLs’ presence in freshwater and groundwater is as extensive as in marine systems [5,6]. Although there are still many questions to be answered on the potential hazard that MNPLs pollution pose to human health, recent studies evidence the potential of MNPLs’ to (1) translocate through physiological barriers and internalize in cells [7,8]; (2) produce cytotoxicity [9]; (3) increase the levels of intracellular reactive oxygen species (ROS) [10]; (4) induce DNA damage [11]; and (5) trigger the altered secretion of pro-inflammatory cytokines [12]. It is noteworthy the lack of exploration of MNPLs effects under long-term exposure conditions, which would be more descriptive of a real environmental exposure scenario.

In the actual context of environmental pollution, diverse contaminants coexist in the same site and, potentially, they can interact with each other. Therefore, increasing interest is being addressed towards the potential role of MNPLs as carriers for other contaminants. MNPLs’ surface features give them the capacity to interact with and adsorb other compounds, such as heavy metals [13,14], or organic pollutants [15,16]. Interestingly, cadmium, titanium, and lead have already been detected in MNPL samples collected in marine ecosystems [17], and traces of arsenic, titanium, nickel, and cadmium have been identified in polyethylene debris from the North Atlantic subtropical gyre [18]. Despite the potential risk derived from the role of MNPLs as carriers for other pollutants, the information on their combined effects is still very limited.

Arsenic is one of the metals potentially sharing environmental compartments with MNPLs. Inherently present in the Earth’s crust, arsenic is part of geological formations widespread worldwide. The weathering of these rocks and minerals releases inorganic forms -arsenate (As^V^) and arsenite (As^III^)- which enter the arsenic cycle as dust or dissolved in water, and are carried by rivers, rain, and groundwater [19,20]. Hence, the intake of contaminated water or food is the main route of exposure to this classic genotoxic and carcinogenic pollutant. Arsenic exposure affects large human populations that are at a higher risk to develop cardiovascular abnormalities, neurological alterations, hepatotoxicity, and most importantly cancer. Being a well-described carcinogen, long-term exposures to arsenic have been associated with lung, bladder, liver, skin, prostate, and kidney cancer [21]. Multiple studies have addressed arsenic-induced carcinogenesis following diverse in vitro, in vivo, and epidemiological approaches. In vitro, 12–30 weeks of chronic arsenic exposure transforms very different cell models, including lung epithelial cells [22], prostate epithelial cells [23], breast epithelial cells [24], keratinocytes [25,26], and fibroblasts [27].

As previously mentioned, the main health hazard associated with arsenic contamination is the intake of contaminated groundwater, which is also potentially contaminated with MNPLs. As a representative example, microplastic fibers have been detected in karstic groundwater systems at a concentration of 15.2 particles/L in certain areas of the USA [28]. There is still limited information regarding the quantification of MNPLs levels in groundwater; however, given the widespread nature of plastic contamination, it seems likely that MNPLs and arsenic are found together in highly polluted areas. An important implication of this coexistence is that MNPLs could act as arsenic carriers, altering their toxicological profile. Further, considering the existing evidence of arsenic adsorption onto MNPLs [29,30], it is very relevant to describe the probable health risk posed by this interaction, which remains unexplored up to date.

In this work, we have determined the interaction between arsenic and PSNPLs, the latter being one of the most abundant MNPLs in the environment [1]. Further, to assess the co-exposure impact under long-term conditions, we have used as a cell model *Ogg1* deficient mouse embryonic fibroblasts, originally transformed by 30 weeks of chronic exposure to sodium arsenite (As^III^), and hereafter referred to as prone-to-transformation progress (PTP) cells. PTP cells show characteristic cancer-associated features, such as morphological changes, differentiation status deregulation, and invasiveness potential [31]. Furthermore, this PTP cell model has been proven useful to demonstrate the carcinogenic potential of different nanomaterials, such as cobalt and zinc oxide [32,33]. Rather than evaluating tumor induction, the transformed status of our selected PTP model allows the evaluation of potential tumor-promoting effects of an MNPLs/As^III^ co-exposure, which might otherwise go unnoticed when standard genotoxicity approaches are used. Thus, aiming to assess whether the chronic PSNPLs exposure, or the PSNPLs/As^III^ co-exposure, exacerbate the transformed phenotype of the PTP cells, they were exposed for 12 weeks to PSNPLs, As^III^, or the combination of both compounds, and changes in different hallmarks of carcinogenesis were evaluated. In all cases, subtoxic concentrations of PSNPLs and As^III^ were used.

## 2. Methods

### 2.1. PSNPLs and y-PSNPLs Characterization

Pristine PSNPLs (PP-008-10) and yellow fluorophore-conjugated PSNPLs (y-PSNPLs) (FP-00552-2) were purchased from Spherotech (Chicago, IL, USA). The pristine form of polystyrene (PS) particles was used for all the experiments carried out except for those in which the fluorescent marker was required, such as the visualization and the quantification of PS particles internalization by confocal microscopy and flow cytometry, respectively. PSNPLs used for the assays were characterized using Zetasizer and transmission electron microscopy (TEM) methodologies. To that purpose, the obtained dispersions were diluted to a final concentration of 100 μg/mL in distilled water. The hydrodynamic size and the Z-potential values for PSNPLs and y-PSNPLs dispersions were determined in a Malvern Zetasizer Nano ZS zen3600 device (Malvern, UK). All the parameters for each sample were measured in triplicates. TEM grids were dipped into PSNPLs and y-PSNPLs dispersions and visualized on a JEOL JEM-1400 instrument (JEOL LTD, Tokyo, Japan). To determine the mean size, 100 randomly selected PSNPLs were measured using an Image J software with the Fiji extension.

### 2.2. Cell Culture Conditions

Mouse embryonic fibroblasts phenotypically sensitive to oxidative damage (*Ogg1^−/−^*), and previously transformed by 30 weeks of continuous sodium arsenite (As^III^) exposure (2 µM) [34], were used in this study and will be henceforth referenced to as prone-to-transformation progress (PTP) cells. PTP cells were grown in DMEM medium supplemented with 10% fetal bovine serum, and 2.5 µg/mL Plasmocin, as previously reported [12].

### 2.3. PSNPLs Uptake by PTP Cells

The cellular localization of y-PSNPLs was determined by confocal microscopy to assess PSNPLs internalization by PTP cells. To this end, 80,000 cells were seeded in Glass Bottom Microwell dishes (MatTek, Ashland, OR, USA) and exposed to 25 and 100 µg/mL y-PSNPLs, for 24 h. The samples were washed with PBS 1X, and nuclei and cell membranes were stained with 1:500 Hoechst 33342 (ThermoFisher Scientific, Carlsbad, CA, USA) and 1:500 Cellmask™ Deep Red plasma (ThermoFisher Scientific, Carlsbad, CA, USA), respectively, for 15 min at room temperature. y-PSNPLs were detected thanks to the fluorophore to which they are conjugated. A Leica TCS SP5 confocal microscope was used to visualize two different randomly selected fields per sample, and images were processed with Image J software with the Fiji extension. In addition, quantification of y-PSNPLs internalization was carried out by flow cytometry. Briefly, after the PTP cells exposure to 25 and 100 µg/mL of y-PSNPLs, cells were washed with PBS 1X, trypsinized, centrifuged, and recovered in PBS 1X at a final concentration of 1 × 10^6^ cells/mL in FACS tubes. To select live cells from the total population of the samples, 1:1000 propidium iodide was added before the analysis with a BD FACSCanto Flow Cytometer (BD Bioscience, Franklin Lakes, CA, USA). About 10,000 living cells per sample were analyzed and the y-PSNPLs uptake was extrapolated from the mean fluorescence intensity of the living cells population. PTC cells incubated with DMEM/F12 medium were used as a control in both assays.

### 2.4. PSNPLs/As^III^ Interaction Detection

The interaction between As^III^ and PSNPLs was assessed by TEM. With this aim, the highest doses of both treatments were used. Concisely, a dilution in distilled water with a final concentration of 20 µM As^III^ and 100 µg/mL PSNPLs was incubated for 3 h at room temperature. TEM grids were dipped into the sample and analyzed by transmission electron microscopy coupled with energy-dispersive X-ray spectroscopy (TEM-EDX). A TEM JEOL-2011 (200 kV) instrument (JEOL LTD, Tokyo, Japan) was used to visualize the sample and take images, while an INCA detector (Oxford Instruments, Abingdon, UK) was used to determine the elementary composition of the sample to detect arsenic on PSNPLs surface.

### 2.5. As^III^ Internalization by PTP Cells

To evaluate and quantify the As^III^ uptake by PTP cells, an analysis with inductively coupled plasma mass spectrometry (ICP-MS) was performed. PTP cells were exposed to 20 µM As^III^ combined with 0, 25, and 100 µg/mL PSNPLs for 24 h. PTP cells non-exposed to As^III^ but exposed to 0, 25, and 100 µg/mL PSNPLs for 24 h were used as negative controls. After that, cells were washed with PBS 1X and trypsinized. Then, samples were centrifuged at 1000 rpm for 8 min, supernatants were discarded, and pellets were frozen at −20 °C until 30 min digestion in concentrated HNO_3_ (Merck, supra pure) on a heat block at 105 °C. Finally, the amount of arsenic in each sample was determined using an ICP-MS 7500-ce device (Agilent Technologies, Santa Clara, CA, USA).

### 2.6. In Vitro Chronic PSNPLs and Arsenic (Co)Exposure

PTP cells were (co)exposed for 12 weeks to 25 µg/mL PSNPLs, 2 µM As^III^, or the combination of both treatments: 25 µg/mL PSNPLs/2 µM As^III^. These concentrations were selected as being within the low range of doses with subtoxic effects (see Appendix A). The combined treatment was prepared 3 h before its addition to the culture medium, mixing PSNPLs and As^III^ in sterile dH_2_O to facilitate the formation of PSNPLs/As^III^ complexes. Replicates of exposed and passage-matched PTP cells were maintained in two separate T-25 flasks and grown under the culture conditions previously described. To ensure a constant exposure condition in the long-term exposure experiments, the cell culture medium was replaced every 2–3 days with new media containing the desired concentration of PSNPLs, As^III^, of the combination PSNPLs/As^III^.

### 2.7. Comet Assay

The total and oxidative DNA damage (ODD) for PTP cells under the different exposure scenarios was evaluated by the comet assay with/without FPG enzyme, as already described [34]. Sheet films of the type Gelbond^®^ (GF) (Life Sciences, Vilnius, Lithuania) were used as a support. Briefly, cells were collected by trypsinization, centrifuged, and resuspended in cold PBS at 17,500 cells/25 μL. Then, cells were mixed with 0.75% LMP agarose at 37 °C (1:10) and 7 μL of the mixture was dropped onto the GF. The GF were lysed overnight by immersion in a lysis buffer (2.5 M NaCl, 0.1 M Na_2_EDTA, 0.1 M Tris Base, 1% Triton X-100, 1% lauryl sarcosinate, 10% DMSO; pH 10). Then, they were gently washed twice in enzyme buffer (10 mM HEPES, 0.1 M KCl, 0.5 mM EDTA, 0.2 mg/mL BSA; pH 8) and incubated in such buffer (negative control) or FPG-containing enzyme buffer. After an electrophoresis buffer washing, GF were submitted to electrophoresis (20 min at 0.8 V/cm and 300 mA) at 4 °C. The GF, rinsed with cold PBS and fixed in absolute ethanol, were stained with SYBR Gold and analyzed according to their percentage of DNA in the tail, as an adequate measure of DNA damage, using semi-automatic software scoring [11].

### 2.8. Cell Morphology and Spindle-like-Cell Proportion Calculation

Morphological changes are associated with cell transformation and result from changes in the network of filaments constituting the cytoskeleton mainly responsible for controlling and maintaining cellular morphology and motility. To qualitatively evaluate cell morphology, cells were photographed with a Zeiss Observer A1 microscope. Quantification of the spindle-like cells was carried out with Image J with the Colocalization Object Counter plugin [35]. Approximately 200 cells were scored for each of the five randomly selected images (10× magnification), total and spindle-like cells were counted and the proportion of spindle-like cells in each field was calculated.

### 2.9. Anchorage-Independent Growth Induction

To assess the cells’ anchorage-independent growth potential, colony formation in soft-agar was determined for PTP cells and the different exposure conditions. The method recently described was used [36]. A suspension of 65,000 cells in 1.75 mL of DMEM containing 10% of FBS and 2.5 µg/mL Plasmocin was prepared and mixed with DMEM containing FBS, NEEA, L-Glu, penicillin-streptomycin, and bacto-agar; 20,000 cells (1.5 mL) were placed in each well of a 6-well plate containing a 0.6% base agar. Plates were maintained at 37 °C for 21 days. To detect the cells’ ability to form colonies they were stained with INT, scanned with an HP Scanjet G4050, and the resulting colonies were counted using the colony cell counter enumerator software OpenCFU (3.9.0).

### 2.10. Invasion and Migration Induction

The invasive potential of PTP cells, and those subjected to further As^III^ and PSNPLs exposures, were evaluated by performing direct migration and invasion assays. To carry out the invasion assay, cells at 80% confluency were deprived of FBS for 24 h. Deprived cells were placed on the apical side of a transwell insert cover with a Matrigel^®^ (Costar-Cornic, Lowell, MA, USA) mixture while the basolateral side contained DMEM complemented with FBS as a chemoattractant medium. Cells were then allowed to invade for 48 h and invading cells were collected and counted.

A modified version of the assay was performed to evaluate cell migration. The main steps were followed as described above; however, the cells were seeded on the top of the transwell without the Matrigel^®^ coating.

### 2.11. Tumorsphere Formation Induction

PTP exposed cells were seeded at a density of 2500 cells/mL on 96-well ultra-low-attachment plates (Corning, Costar-Corning, Lowell, MA, USA) in serum-free DMEM/F12 supplemented with B27, 20 ng/mL basic fibroblast growth factor (bFGF) (both from Life Technologies, Grand Island, NY, USA), epithelial growth factor and 4 µg/mL heparin (both from Sigma-Aldrich, Steinheim, Germany). After 6 days of incubation (5% CO_2_ and 37 °C), the tumorspheres were counted and photographed. Tumorspheres’ size was assessed using ImageJ software.

## 3. Results

### 3.1. PS Materials Characterization

Both, PSNPLs and y-PSNPLs were visualized by TEM. As shown in Figure 1A, the commercial PSNPLs and y-PSNPLs dispersions consist of electrodense round-shaped particles. Particle size was measured from TEM images by Image J, obtaining median sizes of 45.91 nm and 42.42 nm for PSNPLs and y-PSNPLs, respectively (Figure 1B). PS materials were further characterized by Z-sizer and data obtained is summarized in Figure 1B. PSNPLs and y-PSNPLs hydrodynamic radius measured by DLS were larger than those measured from TEM images. Besides, polydispersity index (PdI) values close to 0 indicate the samples are consistently monodispersions, and Z-potential measurements indicate high stability of the dispersions.

### 3.2. Determination and Quantification of y-PSNPLs Uptake by PTP Cells

The cellular location of y-PSNPLs after the internalization by PTP cells was assessed by confocal microscopy. y-PSNPLs were found inside the cell cytoplasm in all conditions analyzed (Figure 2A). However, no difference in the nanoplastic internalization pattern could be deduced at first sight with the tested concentrations. To quantify y-PSNPLs cellular uptake, the mean fluorescence intensity of the living PTP cells exposed to 25 and 100 µg/mL y-PSNPLs was determined by flow cytometry. As shown in Figure 2B, there is a dose-dependent significant increase of the fluorescence intensity, which reflects a greater internalization of y-PSNPLs as the selected doses increase.

### 3.3. Visualization of the PSNPLs/As^III^ Interaction in Dispersion, and Quantification of As^III^ Uptake by PTP Cells

As^III^ and PSNPLs interactions in dispersion were visualized by TEM and the presence of arsenic was confirmed with EDX analysis. As shown in Figure 3A, different types of associations between As^III^ and polystyrene particles were observed. On the one hand, arsenic was found associated with single PSNPLs. Differently, arsenic was also visualized forming aggregates that were, in turn, ringed by PSNPLs. The plot of the arsenic electrons transition shown in the diagrams obtained with EDX confirmed that the electrodense shadow detected with TEM consists of As^III^. Once confirmed the interaction and the formation of PSNPLs/As^III^ complexes, arsenic uptake by PTP cells was determined by ICP-MS. The amount of arsenic detected inside the cells did not show a PSNPLs dependency. As shown in Figure 3B, the amount of internalized arsenic remained unchangeable at the different PSNPLs concentrations assayed. The pg of arsenic measured was normalized to the number of cells in each sample. In the negative control samples, arsenic was under the limit of detection as expected. Since the ICP-MS device was unable to detect it, data are arbitrarily represented as 0.005 pg As^III^/cell.

### 3.4. Both the Co-Exposure and the Single Pollutants Exposure Induce DNA Damage

Given the obtained evidence supporting the interaction between the selected pollutants, we evaluated the genotoxic potential of the chronic (co)exposure to arsenic and PSNPLs. The levels of DNA damage and oxidative DNA damage (ODD) of PTP cells were assessed after 12-weeks of prolonged PSNPLs, As^III^, or PSNPLs/As^III^ exposure. As seen in Figure 4, all exposure scenarios tested led to an increase of the direct (Figure 4A) and oxidative (Figure 4B) DNA damage, compared to the damage levels of unexposed PTP cells. Interestingly, the levels of ODD in As^III^- and PSNPLs/As^III^-exposed PTP cells were also significantly greater than those exposed to PSNPLs alone. This indicates the potential induction of oxidative stress that would affect the DNA bases.

### 3.5. The PTP Oncogenic Phenotype Is Exacerbated by the Long-Term Co-Exposure to As^III^ and PSNPLs

To determine whether the aggressiveness of the oncogenic phenotype of PTP cells is enhanced after the 12-week-extended exposure to As^III^, PSNPLs, or the combination PSNPLs/As^III^, several carcinogenesis biomarkers were evaluated. The cells’ proliferation rate remained unchanged during the weeks of long-term exposure. However, as shown in Figure 5, a certain level of morphological changes is evidenced by the increase in the proportion of spindle-like cells in the culture of PSNPLs- and As^III^-exposed cells, and in those subjected to the co-exposure, in comparison to non-exposed PTP cells.

At a functional level, the soft-agar assay showed a significant 6-fold increase in the number of colonies formed by PTP cells under PSNPLs/As^III^ exposure settings, when compared with passage-matched or single-exposed PTP cells (Figure 6A). Interestingly, no effects of the chronic PSNPLs exposure were observed. Regarding the cells’ migrating (Figure 6B) and invading (Figure 6C) potential, PSNP- and As^III^-exposed cells showed a similar ability to cross the porous membrane and translocate to the basolateral part of the transwell as that of PTP cells, while the number of PSNPLs/As^III^ co-exposed cells able to migrate and invade, was 2-fold and 3-fold higher, respectively.

### 3.6. The Increased Aggressiveness of PTC’s Oncogenic Phenotype Is Not Related to an Increasing Stem-like Cells Population

The cells’ capacity to grow as tumorspheres is associated with the presence of stem or progenitor cells in tumor cell populations. Thus, the number of tumorspheres formed can be used to characterize the cancer stem-like cells within a population of in vitro cultured cancer cells resulting from the applied exposures. When we evaluated if the potential of PTP cells to form tumorspheres was exacerbated by the extended PSNPLs, As^III^, and combined exposures, no differences were found (Figure 7). This would indicate that, under the tested exposure conditions, the pollutants do not induce an increase in the number of cancer stem cells within the PTP cell population.

## 4. Discussions

The widespread and ever-rising amount of plastic litter is closely linked to the increasing levels of MNPLs found in all environmental compartments. These tiny particles (<5 mm) are ubiquitously distributed in soil, air, freshwater, and marine ecosystems from where they easily enter the trophic chain [37]. As a result, humans are believed to be mainly exposed to MNPLs via the ingestion of contaminated food or water, but also through other routes, such as inhalation or dermal deposition [38]. Under this potential broad human exposure scenario, urgent hazard assessments are required.

Great efforts are being addressed to the understanding of the environmental and ecotoxicological effects of these emergent contaminants. However, studies based on mammal and human models focusing on the characterization of the MNPLs’ impact on human health are still limited [39]. Among those available in the literature, it has been fairly described that MNPLs significantly internalize cells and translocate through physiological barriers [7,8]; however, whether this uptake results in a biological impact is not sufficiently clear. While some authors describe a lack of cytotoxic and cytostatic effects [40,41], others have reported MNPL-induced ROS production and pro-inflammatory responses in vitro, as well as mild histological lesions and metabolic disorders in rodent systems [42]. Regardless of these disparities, overall, MNPLs are considered to have low acute toxicity. Nonetheless, much remains to be unveiled in terms of MNPLs’ long-term effects and their role as carriers of different environmental pollutants, which is now attracting attention as a potential toxicological risk associated with MNPLs exposure. 

In this context, our work contributes to the field in two ways: (1) by the analysis of the impact of MNPLs and arsenic (co)exposures under a long-term exposure scenario; and (2) by the establishment of a model in which the already damaged genetic background of cells may render them more susceptible to the alterations induced by MNPLs, allowing the detection of typically unnoticed mild effects.

Our selected co-contaminants of study are As^III^ and PSNPLs, as representative legacy and emergent contaminants, respectively. They share a ubiquitous environmental distribution although both have major implications in terms of human exposure via the intake of contaminated water and, to a lesser proportion, inhalation. As a result, primary target organs (gastrointestinal and respiratory tracts) are potentially affected by both contaminants [43,44]. Thus, these coexisting contaminants could have a joint impact on human health. Indeed, there is accumulating evidence hinting at MNPLs/arsenic interaction. Arsenic adsorption onto MNPLs has already been reported in debris samples collected from the open sea [18]. Besides, laboratory studies have confirmed arsenic adsorption onto polystyrene microplastic particles [30], and polytetrafluoroethylene microparticles [29]. Accordingly, our data demonstrate that arsenic adsorbs onto single PS particles, and it can form PSNPLs/As^III^ aggregates (see Figure 3A). Although the proportion of interactions within our samples is not quantifiable with the use of TEM/EDX, the physical interaction is observed and, thus, the generation of a certain number of PSNPLs/As^III^ complexes could induce differential effects compared to those of the addition of the arsenic and MNPLs as independent compounds.

Aiming to test the long-term impact of the PSNPLs/As^III^ complexes, and to demonstrate whether the effects induced by arsenic exposure are exacerbated under a co-exposure scenario, we have evaluated endpoints regarding genotoxicity and carcinogenicity after the chronic (co)exposure of PTP cells, our selected in vitro model. The used PTP cells derive from MEF cells previously demonstrated to be sensitive to oxidative stress which is closely linked with genotoxicity, genomic and chromosomal instability, and the eventual cell transformation driven by 30-weeks of chronic arsenic exposure [31,34]. We assume the compromised genetic background of these cells can be helpful to make more evident the biological impact of MNPLs and their co-exposures.

The high cellular uptake of PSNPLs in our system (see Figure 2A,B), led us to consider whether the PSNPLs/As^III^ interaction would translate into an increased arsenic bioavailability and a higher internalization rate. This phenomenon has already been described upon in vitro co-exposures to arsenic and nanoparticles (NPs), such as TiO_2_NPs and SiO_2_NPs [45,46], and after the combined exposure to PSNPLs and AgNPs [14]. However, as shown in Figure 3B, the levels of arsenic internalized in PTP cells remained stable with increasing doses of PSNPLs. Therefore, the remarkable genotoxic/oncogenic effects observed upon PSNPLs/As^III^ co-exposure are not due to the PSNPL-mediated facilitation of As^III^ uptake, but rather due to potential alterations induced by PSNPLs/As^III^ at the molecular level.

Among those notable effects of arsenic and MNPLs (co)exposure in our system, we have found a significant induction of both direct and oxidative DNA damage. Arsenic is a well-known genotoxic compound and one of its most studied mechanisms of action is the induction of ROS and oxidative stress [47]. In addition, plenty of those studies reporting the adverse effects of MNPLs have detected increased ROS levels and DNA damage after short-term exposure [10,11,12,48]. Concordantly, with the analysis of the long-term effects of the exposure to PSNPLs, As^III^, and PSNPLs/As^III^ we found a significant increase in the total and oxidative DNA damage when compared with passage-matched PTP cells (see Figure 4). Interestingly, the oxidative DNA damage derived from the PSNPLs/As^III^ co-exposure is significantly higher than that observed after single exposures (see Figure 4B). It should be emphasized the advantage of using the comet assay complemented with the FPG enzyme as a powerful technique to detect not only oxidative stress but their effects on DNA as a target. It should be indicated that FPG primarily exhibits a substrate preference for purines, such as 8-oxoG and FapyG, although oxidized pyrimidines are also removed. Interestingly, the Fpg protein shares similar substrate specificity with the human Ogg1 enzyme. This effect of contaminant mixtures enhancing arsenic-induced oxidative stress and genotoxicity has also been recently reported after short-term exposures when analyzing the impact of the co-exposure to As^III^/TiO_2_NPs [45], As^III^/SiO_2_NPs [46], and As^III^/polystyrene microplastics [49]. Therefore, our results and those from other groups contribute to support the existence of positive interactions between contaminants.

Remarkably, the positive interaction between As^III^ and PSNPLs also adds to the aggressiveness of the arsenic-induced oncogenic phenotype. Arsenic capacity to drive in vitro carcinogenicity is well established [50], while this potential aspect of MNPLs’ long-term impact has not been explored up to date. In in vitro studies, the usefulness of a battery of cancer hallmarks to assess the transformed status of cells has been proposed [51]. These include morphological changes, accelerated proliferation, secretome alterations, metastatic potential, and deregulation of the differentiation status. The measure of these features has been proven useful to assess arsenic-induced carcinogenesis before [52,53,54]. In the present study passage-matched PTP cells that, as previously mentioned, meet all endpoints, display an evident transformed phenotype [31]. Interestingly, according to our data, these hallmarks remain unchanged for PTP cells subjected to 12 weeks of PSNPLs or As^III^ single exposure but are significantly enhanced under PSNPLs/As^III^ co-exposure settings. The increment in the proportion of spindle-like cells within the population (see Figure 5A,B), and especially the dramatically increased capacity of cells to grow independently of anchorage (see Figure 6A), migrate (see Figure 6B), and invade (see Figure 6C) confirm the co-exposure-mediated acquisition of a further aggressive transformed phenotype. To better characterize the oncogenic features of the (co)exposed PTP cells, we evaluated their tumorsphere-forming ability as a marker of the stemness status in our cell population. Different studies have linked the conversion of non-stem cells to cancer stem cells with carcinogenesis [55], and, specifically, with arsenic-induced carcinogenesis [56,57,58,59]. However, we did not find significant differences under the different exposure scenarios tested, thus stemness induction seems not to be the mechanism by which PSNPLs/As^III^ promotes tumor aggressiveness. Taken together, these findings highlight the urgent need to explore MNPLs’ long-term effects and their potential role as co-carcinogens with other environmental pollutants.

## 5. Conclusions

As a summary, in the present work, we have demonstrated that the long-term concurrent exposure to subtoxic doses of arsenic and PSNPLs significantly enhances the arsenic-associated transformed phenotype and genotoxicity. Further, we have shown that PSNPLs and arsenic physically interact. However, this interaction is not associated with increased arsenic bioavailability and, therefore, the mechanism by which PSNPLs add to arsenic’s impact requires further research. Importantly, while the evaluation of the effects induced by a single exposure to MNPLs is still of great relevance, it is also necessary to explore whether new differential effects arise with environmentally relevant complex mixtures. Indeed, our results support considering co-exposure scenarios as an essential part of the emergent pollutants’ hazard assessment.

## Figures and Tables

**Figure 1 ijms-23-02958-f001:**
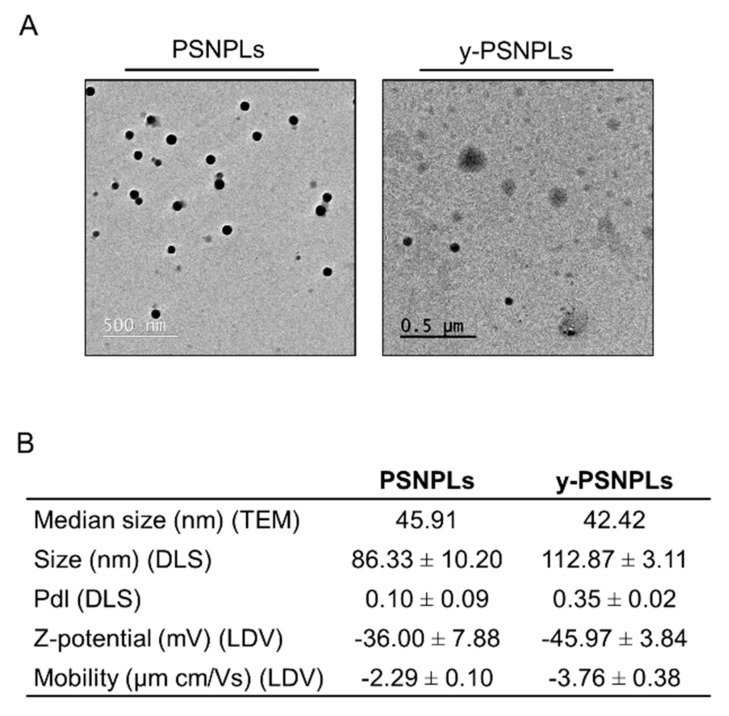
PS nanomaterials characterization. (**A**) Representative TEM images of PSNPLs and y-PSNPL. (**B**) PSNPLs and y-PSNPLs characterization by TEM (median size calculated measuring 100 randomly selected PSNPLs) and Zetasizer Nano ZS (mean ± SD). 100 μg/mL dilutions in distilled water of each material were used for the visualization and characterization.

**Figure 2 ijms-23-02958-f002:**
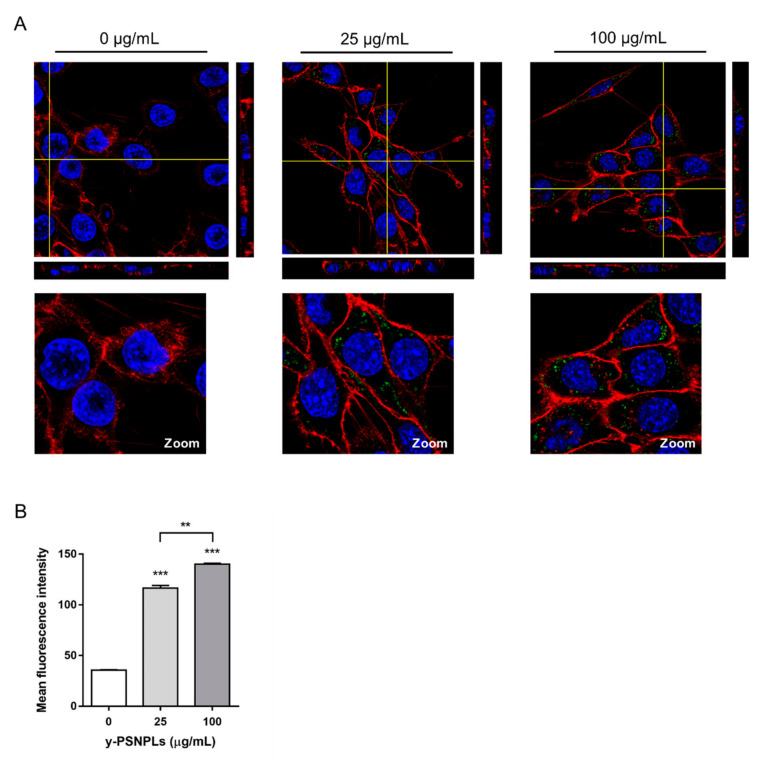
y-PSNPLs uptake by PTP cells. (**A**) Images of the PTP cells were taken with confocal microscopy after the exposure to 0, 25, and 100 µg/mL y-PSNPLs for 24 h. Nuclei (showed in blue) were stained with Hoechst and cell membranes (showed in red) were stained with CellMask. y-PSNPLs are shown in green. Yellow lines point out the plane from where orthogonal views are projected. (**B**) y-PSNPLs intake by PTP cells after 24 h of exposure to 0, 25, and 100 µg/mL. The mean fluorescence intensity of the living PTP cells’ total population is represented. Data are shown as mean ± SEM and analyzed by the Student’s *t*-test (*** *p* < 0.001, ** *p* < 0.01, compared to non-exposed PTP cells controls or other conditions as indicated).

**Figure 3 ijms-23-02958-f003:**
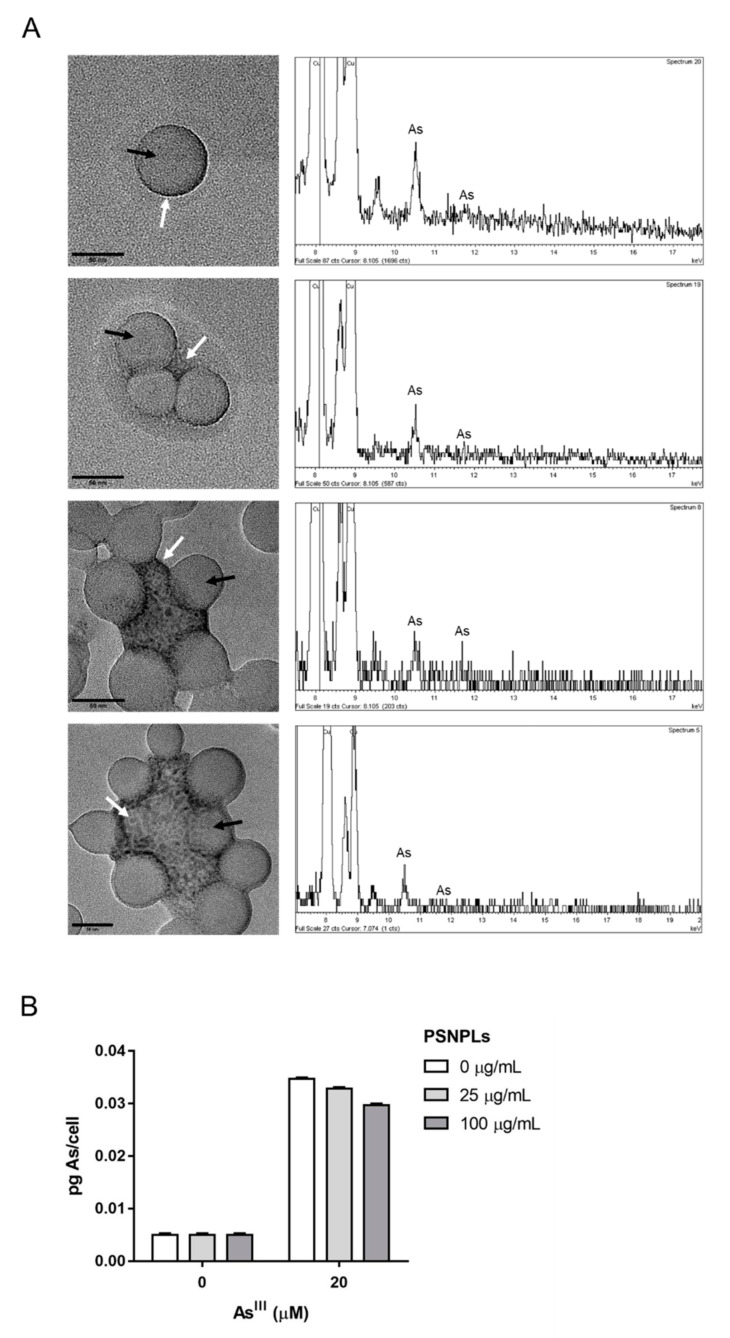
Determination of the interaction between PSNPLs and As^III^, and the As^III^ uptake by PTP cells. (**A**) Representative TEM images with their respective energy-dispersive X-ray EDX spectra indicate the chemical elemental characterization. Arrows point to the round-shaped PSNPLs (black) or the areas where arsenic was detected (white) (**B**) Quantification of the arsenic internalization by PTP cells. Arsenic was not detected in control samples; thus, results are arbitrarily represented as 0.005 pg of As^III^. Data are represented as mean ± SEM and was analyzed by the Student’s *t*-test.

**Figure 4 ijms-23-02958-f004:**
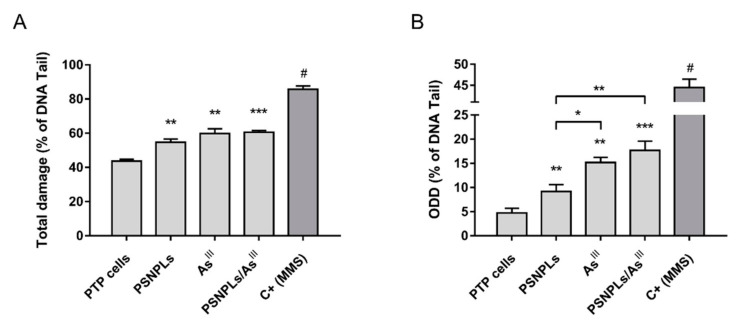
DNA damage induced by the long-term (co)exposure to As^III^ and PSNPLs measured by the comet assay. (**A**) Total and (**B**) oxidative DNA damage levels after a 12-week exposure of PTP cells to 25 µg/mL PSNPLs, 2 µM As^III,^ and the combination of both PSNPLs/As^III^ treatments. Methyl methanesulfonate (MMS; 200 μM) was used as a positive control. Data are presented as mean ± SEM analyzed by one-way ANOVA with Dunnett’s post-test (*** *p* < 0.001, ** *p* < 0.01, * *p* < 0.05 compared to passage-matched PTP cells or other conditions as indicated; ^#^
*p* < 0.001 when compared to all tested conditions).

**Figure 5 ijms-23-02958-f005:**
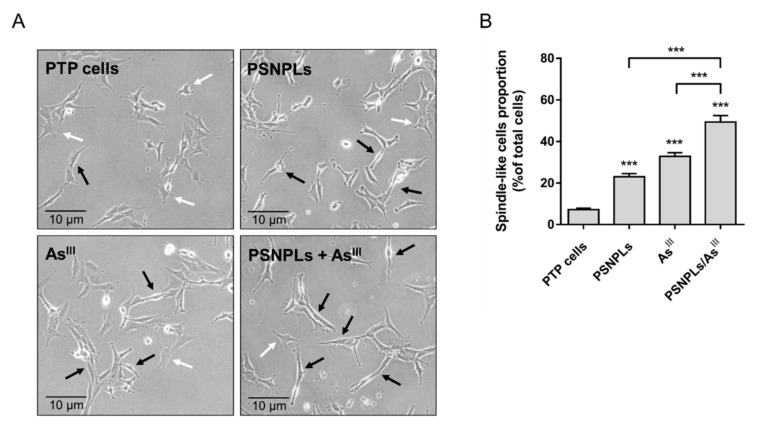
Chronically (co)exposed PTP cells morphology evaluation. (**A**) Representative images of non-exposed PTP, and PTP cells after the chronic exposure to 25 µg/mL PSNPLs, 2 µM As^III^, and the combination of both PSNPLs/As^III^. Arrows point to cells with flattened stellate shape (white) or to spindle-like cells (black). (**B**) The proportion of spindle-like cells was quantified in five different fields for each tested condition. Data are presented as mean ± SEM analyzed by one-way ANOVA with Dunnett’s post-test (*** *p* < 0.001, compared to passage-matched PTP cells or other conditions as indicated).

**Figure 6 ijms-23-02958-f006:**
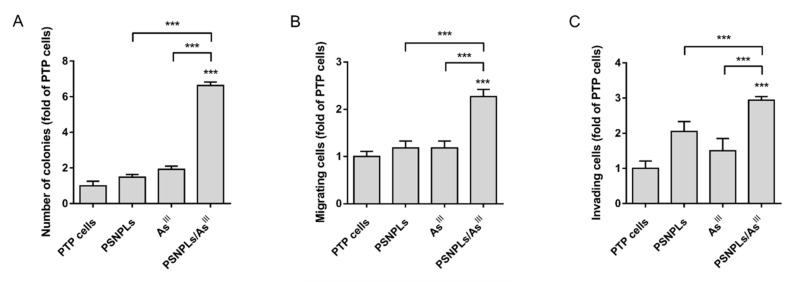
Determination of the in vitro transformed phenotype after the chronic (co)exposure. (**A**) Quantitative data was derived from the number of colonies formed in the anchorage-independent growth assay. (**B**) The proportion of cells able to translocate to the basolateral side of the transwell in the migration, and (**C**) invasion assays. Data are represented as fold of the mean comparing exposed PTP cells with the passage-matched PTP cells. Error bars represent SEM. *** *p* < 0.001 when compared to all conditions tested by the one-way ANOVA with Dunnett’s post-test.

**Figure 7 ijms-23-02958-f007:**
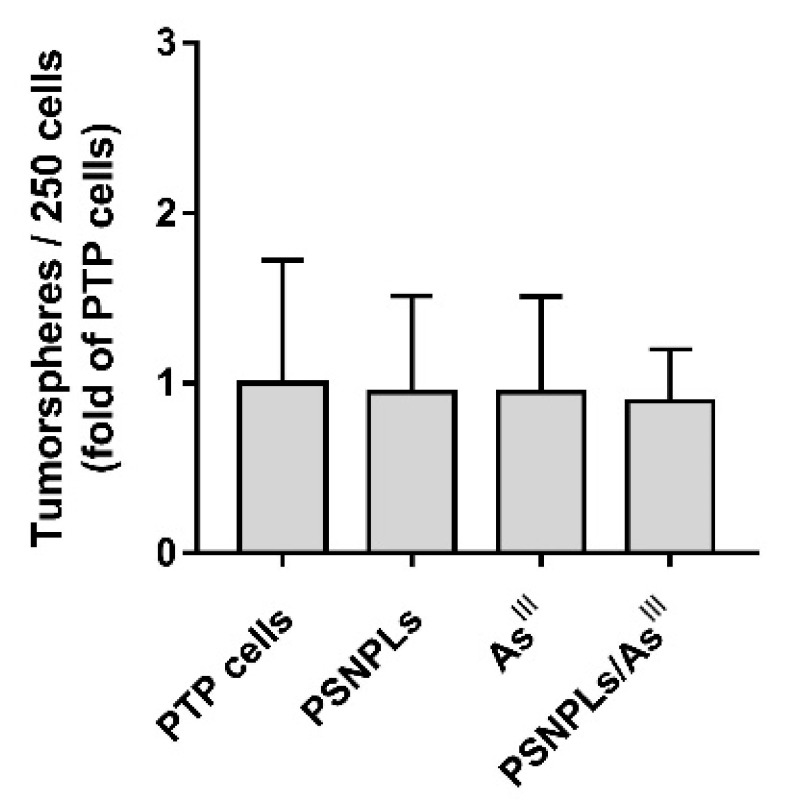
Evaluation of the proportion of stem-like cells in chronically (co)exposed cultures. Several tumorspheres formed for every 250 PTP cells seeded under tumorsphere-inducing conditions. Data are represented as fold of the mean comparing exposed PTP cells with the passage-matched PTP cells. Error bars represent SEM.

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
