# Peer review of "Nanoplastics and Arsenic Co-Exposures Exacerbate Oncogenic Biomarkers under an In Vitro Long-Term Exposure Scenario"

_ijms, 2022, doi:10.3390/ijms23062958_

Round 1

Reviewer 1 Report

This study aims to investigate the potential carcinogenic effects of the nanoplastics and arsenic and the combination of both compounds in prone-to-transformation progress (PTP) cells exposed for a long time (e.g., 12 W). The question appears interesting, but I have a doubt about the approach used to perform the experiments.

The authors declare that they will use a cell model Ogg1 deficient mouse embryonic fibroblasts, originally transformed by 30 weeks of chronic exposure to sodium arsenite (AsIII), here namely PTP, to demonstrate the carcinogenic potential of the polystyrene nanoplastics and arsenic. However, all the experiments show a control that they name AsTC, meaning non- transformed MEF cells or PTP?  This is a focal point that needs to be clarified. However, I think that showing results of both transformed and non-transformed cells is mandatory for respecting the standard parameter of quality of the IJMS.

As minor, I would like to suggest:

- introducing in the abstract the type of cells used for the study

- Consider revising some keywords putting here words that are not nominated in the text or in the title

- If AsTC means PTP, putting in the figure PTP instead of AsTC

- specifying the lyses buffer composition (2.7. Comet assay)

- indicating in the figure 3A what is As and what is PS

- specifying which cells have levels of ODD greater than those from PSNPLs-exposed PTP cells

- controlling Figure 5. I think that there is a mistake. The authors write “ Representative images of non- transformed MEF cells, non-exposed PTP, and PTP cells after the chronic exposure to 25 μg/mL PSNPLs, 2 μM AsIII, and the combination of both PSNPLs/AsIII” but the figures are 4 and not 5.  

I think that taking up this suggestion the manuscript would be enriched with interesting and reliable information for a future submission.

Author Response

RESPONSE TO REVIEWERS

This study aims to investigate the potential carcinogenic effects of the nanoplastics and arsenic and the combination of both compounds in prone-to-transformation progress (PTP) cells exposed for a long time (e.g., 12 W). The question appears interesting, but I have a doubt about the approach used to perform the experiments.

The authors declare that they will use a cell model Ogg1 deficient mouse embryonic fibroblasts, originally transformed by 30 weeks of chronic exposure to sodium arsenite (AsIII), here namely PTP, to demonstrate the carcinogenic potential of the polystyrene nanoplastics and arsenic. However, all the experiments show a control that they name AsTC, meaning non- transformed MEF cells or PTP?  This is a focal point that needs to be clarified. However, I think that showing results of both transformed and non-transformed cells is mandatory for respecting the standard parameter of quality of the IJMS.

RESPONSE:

The figures have been corrected to specify that the passage-matched control cells are PTP cells. We agree with the reviewer on the interest of analysing the different endpoints in non-transformed cells, however the aim of this work is to evaluate the tumor-promoting effects of nanoplastic and the co-exposure to nanoplastic and arsenic. The use of a pre-transformed allows us to detect this effect after shorter exposure times that those that would be required for tumor initiation in non-transformed cells, which nonetheless would be an interesting follow up work once found the significant effects induced by plastic regarding the oncogenic-promoting potential. Accordingly, our results must be considered as an alarm signal about the potential tumoral risk of nanoplastic exposure. Consequently, further studies considering different variables (cell lines, exposure conditions, biomarkers, MNPLs characteristics, …) should be conducted to decipher the potential carcinogenic risk of MNPL exposures.

As minor, I would like to suggest:

- introducing in the abstract the type of cells used for the study

RESPONSE:

The suggestion has been included in the abstract, which now reads as follows: “Mouse embryonic fibroblasts deficient for oxidative DNA damage repair mechanisms and showing a cell transformation status were used as a sensitive cell model”.

- Consider revising some keywords putting here words that are not nominated in the text or in the title

RESPONSE:

The keyword “genotoxic DNA damage” has been changed to “DNA damage” which refers both to genotoxic and oxidative DNA damage and is often mentioned in the main text.

- If AsTC means PTP, putting in the figure PTP instead of AsTC

RESPONSE:

We are sorry for the mistake. As explained previously, the figures have been modified in this respect. We thank the reviewer for this suggestion which indeed clarifies the content of the manuscript.

- specifying the lyses buffer composition (2.7. Comet assay)

RESPONSE:

The composition of the lysis buffer has been included in the method section (lines 207 – 210) which now read as follows: “The GF were lysed overnight by immersion in a lysis buffer (2.5 M NaCl, 0.1M Na2EDTA, 0.1M Tris Base, 1% Triton X-100, 1% lauryl sarcosinate, 10% DMSO; pH 10). Then, they were gently washed twice in enzyme buffer (10 mM HEPES, 0.1 M KCl, 0.5 mM EDTA, 0.2 mg/mL BSA; pH 8) and incubated in such buffer (negative control) or in FPG-containing enzyme buffer.”

- indicating in the figure 3A what is As and what is PS

RESPONSE:

Arrows to indicate the PSNPL particles and the region where arsenic was detected by EDX have been included in the figure and the corresponding description has been added to the figure legend (lines 702-703).

- specifying which cells have levels of ODD greater than those from PSNPLs-exposed PTP cells

RESPONSE:

We agree with the reviewer that the sentence in the originally submitted text was incomplete and, therefore have corrected it. Now the manuscript reads “Interestingly, the levels of ODD in AsIII- and PSNPLs/AsIII-exposed PTP cells were also significantly greater than those exposed to PSNPLs alone.”

- controlling Figure 5. I think that there is a mistake. The authors write “Representative images of non- transformed MEF cells, non-exposed PTP, and PTP cells after the chronic exposure to 25 μg/mL PSNPLs, 2 μM AsIII, and the combination of both PSNPLs/AsIII” but the figures are 4 and not 5.  

RESPONSE:

We thank the reviewer for noticing this mistake which referred to an earlier version of the figure. We have modified the figure legend to match the final figure included in the manuscript as follows: “Representative images of non-exposed PTP, and PTP cells after the chronic exposure to 25 µg/mL PSNPLs, 2 µM AsIII, and the combination of both PSNPLs/AsIII.”.

I think that taking up this suggestion the manuscript would be enriched with interesting and reliable information for a future submission.

Reviewer 2 Report

The submitted study describes the effects of arsenic and polystyrene particle exposure on transformation in a specific cell testing system, the Ogg-deficient murine embryonic fibroblast model. The authors reported increased anchorage-independent growth, enhanced migration and invasion and morphology changes upon co-exposure of As and particles compared to single treatments. This appeared to be due to increased particle but not increased intracellular arsenic uptake. Information is missing to evaluate the significance of the reported data.

-In section 2.6. it is stated that experiments were conducted at subtoxic levels. It would be good to show the cytotoxicity data for the mono- and co-exposures.

-Interaction of arsenic and particles is expected and has been reported in the study. However, it is appears that these studies were performed in distilled water and the interaction may be different upon exposure in cell culture medium with FBS. It would be helpful to show the interaction also in the cell culture medium.

- Can some information be provided on the relevance of the applied doses (realistic?)?

- Data about the predictive value or validation of the used model should be shown.

-Almost all assays lack a positive control, which should be included as usual for cellular assays.

-Does the medium (section 2.2.) contain also other ingredients, e.g. antibiotics, FBS in 10% concentration?

-It is stated in section 2.8.: “For 5 randomly selected images, total and spindle-like cells were counted.” I would be more informative to indicate the number of cells, which were evaluated, because no magnification has been given.

-The demonstration that there is intracellular ROS production should be added (e.g. DCF assay). Maybe there are other mechanisms involved.

Minor

There are a couple of spelling errors, etc. that should be corrected. E.g. “..,this PTP cell model has demonstrate to be..”; “…PTP model allow to examine…”

Author Response

see the attached document

Round 2

Reviewer 1 Report

The manuscript is now ready to be published. Please have a look at the text to remove a few small errors of distraction.

Author Response

We appreciate the positive feedback of the reviewer to our comments/answers.

We have carefully checked the manuscript and the few detected small errors have been corrected

Reviewer 2 Report

Most of my comments were addressed and there are only minor issues left, such as the mention about the 200 cells that were evaluated in section 2.8. and the inclusion of a scale bar (only indicating the original magnification may not be informative because it depends on the image size). 
The dose-dependent cytotoxicity of MP and As could be provided as Supplementary Materials
It needs also to be mentioned as a limitation - if I interpret the reply correctly - that the model has not been validated with conventional compounds (e.g. As is only the inducer and no other chemicals have been tested so far). 

Author Response

Most of my comments were addressed and there are only minor issues left, such as the mention about the 200 cells that were evaluated in section 2.8. and the inclusion of a scale bar (only indicating the original magnification may not be informative because it depends on the image size). 

RESPONSE

We have now included the number of cells (200) scored for each one of the five selected images.

We wish to indicate that the scale bar is enlarged/diminished according to the image size

The dose-dependent cytotoxicity of MP and As could be provided as Supplementary Materials
It needs also to be mentioned as a limitation - if I interpret the reply correctly - that the model has not been validated with conventional compounds (e.g. As is only the inducer and no other chemicals have been tested so far). 

RESPONSE

We have included the dose-dependent cytotoxicity figures as Supplementary Material.

In addition to the initial study using arsenic (references 31 and 34), the model was also successfully tested using cobalt and zinc oxide nanoparticles (see references 32 and 33). In the present study, arsenic itself acts as a positive control, taking into account its well-known carcinogenic potential and the previous positive reported findings with our model.